# Circulating Biomarkers as Potential Risk Factors for Inguinal Hernia

**DOI:** 10.3390/ijms26157032

**Published:** 2025-07-22

**Authors:** Enke Baldini, Salvatore Sorrenti, Eleonora Lori, Luigi Palla, Silvia Cardarelli, Daniele Pironi, Domenico Tripodi, Antonio Pavan, Azis Fakeri, Vilma Cobo, Chiara Pellegrini, Priscilla Nardi, Valerio Rinaldi, Salvatore Ulisse, Piergaspare Palumbo

**Affiliations:** 1Department of Surgery, “Sapienza” University of Rome, 00185 Rome, Italy; salvatore.sorrenti@uniroma1.it (S.S.); silvia.cardarelli@uniroma1.it (S.C.); daniele.pironi@uniroma1.it (D.P.); domenico.tripodi@uniroma1.it (D.T.); pellegrini.1795942@studenti.uniroma1.it (C.P.); priscilla.nardi@uniroma1.it (P.N.); valerio.rinaldi@uniroma1.it (V.R.); salvatore.ulisse@uniroma1.it (S.U.); piergaspare.palumbo@uniroma1.it (P.P.); 2Department of Public Health and Infectious Diseases, “Sapienza” University of Rome, 00185 Rome, Italy; luigi.palla@uniroma1.it; 3Immunohaematology and Transfusion Center, Umberto I General Hospital, “Sapienza” University of Rome, 00185 Rome, Italy; antonio.pavan@uniroma1.it (A.P.); a.fakeri@policlinicoumberto1.it (A.F.); vilma.cobo@uniroma1.it (V.C.)

**Keywords:** inguinal hernia, collagen, metalloproteinases, lysyl oxidase, N-terminal propeptides of type I (PINP) and type III (PIIINP) procollagens

## Abstract

Independent studies reported metabolic alterations in connective tissues of hernia patients, especially involving collagen fibers, compared to healthy controls. In the present work, we evaluated plasma concentrations of metalloproteinases (MMPs) and lysyl oxidase (LOX), enzymes involved in collagen metabolism, and peptides produced during collagen biosynthesis (PINP, PIIINP, and PIVNP) as potential biomarkers for the estimation of hernia risk. Zymography and ELISA assays were performed with plasma samples of 51 patients with primary or recurrent inguinal hernia and 42 healthy controls. A reduction in PINP (*p* = 0.007) and a concomitant increase in PIIINP (*p* < 0.001) were observed in patients. In controls, PINP levels were inversely related to age, whereas in patients PIIINP levels increased with age. Body mass index (BMI) showed a strong positive correlation with PIIINP plasma levels in controls but not in patients (*p* < 0.001). Moreover, patients with larger lesions had the lowest PINP/PIIINP ratio (*p* = 0.003). PIVNP collagen did not differ between controls and hernia patients. Plasma MMP-9 was reduced in patients (*p* = 0.015), while MMP-2 and LOX were unchanged. However, MMP-2 concentrations appeared lower in patients with familial history of hernia compared to those without. In regression analysis, the PINP/PIIINP ratio was inversely related to hernia risk, and a cut-off value of 0.948 was found by ROC analysis which classified hernia patients with a sensitivity of 82.9% and a specificity of 77.1%. In conclusion, our findings identified the PINP/PIIINP ratio as the most relevant molecular predictor of inguinal hernia risk.

## 1. Introduction

Inguinal hernia is a protrusion of abdominal contents through a weak area in the lower abdominal wall. It is quite commonly encountered in the general population, especially in men, and repair interventions represent one of the most frequent surgical practices [1].

A recent systematic review and meta-analysis of 10 population-based studies comprising more than 50 million individuals estimated that the global prevalence of inguinal hernia is 7.7%, with a substantial difference between genders (9.6% in males and 1.3% in females) [2].

Some patient-related factors are known to enhance the risk of inguinal hernia, such as inheritance, patent processus vaginalis, older age, low body mass index (BMI), prostatectomy history, and increased intra-abdominal pressure [3,4]. Other putative risk factors, e.g., chronic constipation, prostatic hypertrophy, smoking habit, or cumulative exposure to daily lifting and standing/walking, have been considered but not yet ascertained. On the whole, the etiology of the hernia is likely attributable to one or more triggers acting on congenitally predisposed individuals.

Over the years, some studies have described metabolic alterations in connective tissues entailing a weakening of the abdominal wall and consequent susceptibility to hernia development [5,6,7,8]. The first observations came out in the 1970s, when lower rectus sheaths of adult men with direct inguinal hernia were found to weigh less than those of healthy controls, while patients with indirect or mixed hernias had intermediate values [9,10]. Further investigations evidenced a higher elasticity and maximal distension, and lower total amount of collagen but greater amounts of extractable collagen and elastic fibers, in the fascia transversalis of patients with direct inguinal hernia compared to controls [11].

The biomechanical strength of connective tissues relies on an extracellular protein scaffold of which collagen fibrils are essential elements [12]. Fibrillar type I collagen accounts for about 90% of total collagen. It is ubiquitously expressed in the extracellular matrix (ECM) and represents the main component in mature tendons and ligaments, where it provides strong resistance to tensile stress [13,14]. Type III collagen is the second most abundant fibril-forming collagen, mainly located in elastic tissues and distensible organs such as embryonic skin, lung, blood vessels, vocal cords, bladder, and uterus. Fibrils containing both type III and type I collagens have smaller diameter and reduced stiffness compared to pure type I collagen fibers, and the incorporation of variable amounts of collagen III in collagen I fibers is thought to regulate mechanical properties of tissues [15].

Independent studies described a reduced collagen I/III ratio in the ECM of the rectus sheath and skin of inguinal hernia patients [8,16,17,18]. Moreover, alterations of collagen turnover biomarkers have been detected in the patient bloodstream, such as the N-terminal propeptides of type I (PINP) and type III (PIIINP) procollagens, released during interstitial collagen formation, the N-terminal propeptide of type IV procollagen (PIVNP), derived from basement membrane collagen, and peptides resulting from collagen degradation [5,19].

To date, the biochemical processes involved in the alteration of tissue collagen turnover remain largely unknown. One study reported significantly lower expression of the enzyme lysyl oxidase-like 1 (LOXL-1), which catalyzes the oxidative deamination of lysine and hydroxylysine residues in collagen and elastin, allowing for the formation of covalent cross-linkages in the transversalis fascia of patients with direct inguinal hernia compared to those with indirect hernia and healthy controls [20]. In the study, reduced levels of tropoelastin and higher levels of elastase were also observed in hernia patients [20].

The breakdown of ECM components has been investigated much more extensively in the pathogenesis of abdominal hernia. Most works focused on matrix metalloproteinases (MMPs), key enzymes in ECM remodeling endowed with collagenolytic activity. Various studies consistently indicated an increase in MMP-2 in the serum and/or tissue levels of hernia patients, especially those having direct, recurrent, or bilateral hernia [21,22,23,24]. For other MMPs, the findings produced so far are scarce, heterogeneous, and mainly obtained on small numbers of subjects. In particular, MMP-9 levels in serum and/or tissues were found increased [25,26], unvaried [19], lower [27,28], or higher in tissues but lower in plasma of hernia patients compared to controls [29]. Such discrepancies likely depend on differences in patient selection, investigation methods, statistical approaches, and possibly confounding variables or effect modifiers that were not taken into account.

The present study aimed to evaluate plasma concentrations of molecules involved in collagen turnover as potential biomarkers associated to inguinal hernia and for predicting hernia risk. In particular, we considered the enzyme lysyl oxidase (LOX), involved in the formation of crosslinks in collagen and elastin; the propeptides PINP, PIIINP, and PIVNP; and the gelatinases MMP-2 and MMP-9. Moreover, the way some patient-related factors may moderate the association between circulating levels of these molecules and inguinal hernia was investigated.

## 2. Results

We initially assessed whether potential risk factors for inguinal hernia (i.e., weightlifting, tobacco smoke, benign prostate hypertrophy, and overweight/obesity) had different frequencies in patients versus controls. Statistical results demonstrated that none of these factors was significantly associated with the patient group, nor with hernia characteristics such as mono- or bilaterality, direct or indirect lesion, and size.

Then, plasma concentrations of all the molecules under study were compared between controls and patients. Zymographic analyses evidenced that both gelatinases were present in plasma almost exclusively in the form of zymogens (pro-MMPs), which were activated inside gels during the experimental procedure (see Figure 1A). Controls and patients showed similar enzymatic activity of MMP-2, while that of MMP-9 was significantly reduced in patients (see Figure 1B). These observations were corroborated by a quantitative immunoassay in randomly assembled subgroups of control and patient samples, which produced analogous results (see Figure 1C).

Moreover, patients showed significantly reduced plasma levels of PINP and increased plasma levels of PIIINP compared to controls; hence, the PINP/PIIINP ratio was considerably lowered in patients (see Figure 2A,B).

On the contrary, the PIVNP levels did not differ between controls and patients. Plasma concentrations of LOX were generally low and, in some cases, undetectable. No significant variation was found for this protein in patients compared to controls (see Figure 3A).

Subsequently, we sought to establish whether circulating levels of the molecules considered could be altered by pathophysiological factors other than hernia disease. Both in control and patient groups, the results obtained indicated that no molecules were affected by weightlifting or the presence of benign prostate hypertrophy. Among healthy subjects, smokers had lower concentrations of LOX compared to non-smokers (*p* = 0.030), and the same was observed in the patient group (*p* = 0.046). When controls and patients were combined, this difference became more significant (*p* = 0.005) (see Figure 3B). Of note, plasma concentrations of LOX were not really changed between smokers and non-smokers (0.059 ± 0.009 vs. 0.056 ± 0.007 ng/mL), but the latter had a higher frequency of non-measurable LOX circulating levels (*p* = 0.011).

Considering continuous BMI, we found a strong monotonic relationship with PIIINP plasma levels in healthy individuals but not in patients (Spearman’s correlation coefficient 0.681, *p* < 0.001) (see Figure 4). Consistently with this result, in the control group, being overweight (BMI ≥ 25) was significantly associated with higher plasma levels of PIIINP, but this association was lost in the patient group (see Figure 4B). Average-weight controls exhibited lower PIIINP levels than both overweight/obese controls and patients of any weight, whereas overweight/obese controls had PIIINP values comparable to those of all patients (see Figure 4B).

The age of control subjects was inversely related only to circulating PINP levels (Spearman’s correlation coefficient −0.522, *p* < 0.001). Such a difference was not evident in the patient group (see Figure 5A). Vice versa, plasma concentrations of PIIINP positively correlated with age in patients (Spearman’s correlation coefficient 0.453, *p* = 0.004) but not in controls (see Figure 5B).

Next, we investigated possible relationships between plasma concentrations of the molecules tested and specific patients’ parameters, namely, familiarity for hernia, type (bilateral, direct, or indirect), and size of the lesion. No differences were noted for bilateral versus unilateral hernias, nor for direct versus indirect hernias. However, patients with size 3 hernias had significantly lower levels of PINP and PINP/PIIINP ratio (see Figure 6).

Interestingly, patients with familial history of the disease showed significantly lower levels of MMP-2 activity (see Figure 7). Based on this finding, we compared both zymographic MMP-2 activity and MMP-2 concentration measured by immunoassay in controls and patients without familial history of hernia, but no variations emerged.

Finally, we performed logistic regressions to estimate the associations between relevant biomarkers and hernia adjusting for the confounders age and BMI. From this analysis, PIIINP emerged as an independent positively associated risk factor for hernia (OR: 1.005; 95% CI: 1.003–1.008; *p* < 0.001). On the other hand, both PINP/PIIINP ratio and BMI were inversely related to the likelihood of hernia onset, and the PINP/PIIINP ratio had a more highly significant relationship with the pathology than PIIINP (see Table 1).

Tentative application of a predictive approach based on PINP/PIIINP ratio values led to an area under the curve (AUC) of 0.873 (CI 95%: 0.794–0.952; asymptotic *p* < 0.001), and the best cut-off was 0.948, discriminating individuals affected by inguinal hernia with a sensitivity of 82.9% and a specificity of 77.1% (see Figure 8).

## 3. Discussion

The main aim of the present study was to evaluate plasma concentrations of molecules involved in collagen turnover as potential biomarkers for estimating hernia risk. Specifically, we considered the enzyme LOX, involved in the formation of crosslinks in collagen and elastin, the propeptides PINP, PIIINP, and PIVNP, and the two gelatinases MMP-2 and MMP-9.

As regards MMPs, the pathogenetic role of alterations found at plasma and/or tissue level in hernia patients is not easy to grasp. It may be speculated that both excess and deficiency of collagenolytic enzymes could lead to structural weakening of connective tissues. In fact, it can be supposed that an increased secretion of MMPs in the extracellular environment accelerates collagen degradation processes making fascial tissues thinner. However, MMPs also have indirect effects on the ECM dynamics through regulation of several enzymatic activities, including MMPs themselves. Therefore, it is possible that a reduced production of MMPs impairs the functioning of enzymes involved in maturation and assembly of collagens and other fibril-forming proteins (i.e., elastin and fibronectin), causing abnormalities that compromise tissue strength.

Our results concerning MMP-2 are not in agreement with those reported in previous studies showing higher MMP-2 serum levels in hernia patients [21,23,25]. Such discrepancy may depend on the small numbers of patients with bilateral, recurrent, or direct hernias in our cohort, in whom the increases in circulating MMP-2 compared to healthy controls were found to be more relevant. Moreover, zymography is a semiquantitative technique that has the advantage of distinguishing active MMPs from zymogens but could be less sensitive than ELISA in detecting small changes in protein concentration. Of note, we observed higher levels of MMP-2 in patients without familial history of groin hernia. This evidence, never described before, indicates that familiarity for hernia may represent a confounder that should be taken into account when evaluating circulating MMP-2 as a potential biomarker or prognostic factor. On the other hand, MMP-9 plasma levels were reduced in hernia patients compared to controls, and this result is in line with previous findings [27,28]. However, MMP-9 did not emerge as an independent statistically significant predictor of hernia risk in our study.

Blood measurements of LOX did not vary significantly between controls and patients. It cannot be excluded, however, that changes in LOX concentration occur in patients at the tissue levels, and it would be worth carrying out further studies to verify this possibility.

So far, only two studies have investigated circulating N-terminal peptides derived from procollagen processing in inguinal hernia patients. The first one reported a significant reduction in PINP in patients with single or multiple hernias, and an increase in PIVNP in patients with unilateral hernia compared to controls [5]. The second study documented lower PINP concentrations in patients with indirect, direct, or recurrent hernias, and lower PIIINP concentrations in patients with primary hernias but not in those with recurrent hernias. Applying the receiver operating characteristic (ROC) curve analysis, these authors indicated the PINP/PIIINP ratio as a potential predictor of recurrent inguinal hernias [19]. By examining the same collagen markers in our case study, we identified age and BMI as potential confounding factors as related both to the risk of inguinal hernia and to PINP and PIIINP plasma levels.

Circulating PINP concentrations are thought to depend mainly on its release from bones, with soft tissues contributing only to a minor extent. Since production of new bone ECM is directly proportional to the growth rate of individuals, plasma levels of PINP are maximal during developmental age and decline in adults [30]. Conversely, a recent work performed on 386 individuals ranging from 20 to 98 yrs of age found small differences in mean PIIINP plasma concentrations among subjects, not relevant enough to warrant a reference age range partitioning [31]. These data, although requiring confirmation from larger studies, suggest that the PINP/PIIINP ratio tends to naturally decrease over the years. Our results are in line with this trend only for the control group, in which plasma PINP was inversely related to age while PIIINP was unchanged. In contrast, in the patient group an opposite pattern emerged in that PINP levels did not vary while PIIIINP levels increased with age. Basically, the negative correlation of the PINP/PIIINP ratio with age was maintained in both groups, but while in controls it depended on the decrease in PINP, in patients it was rather due to the increase in PIIINP with advancing age. Furthermore, PIIINP showed a positive correlation with BMI in the control group, while in patients PIIINP levels were similar to those of overweight/obese controls. Weight gain generates a plethora of stimuli in many tissues, including mechanical stress, production of reactive oxygen species, inflammation, and hypoxia, resulting in maladaptive ECM remodeling in both murine models and humans [32,33]. Based on our observations, PIIINP appears to be a plasma marker indicative of such alterations in control subjects, but not in patients. To date, there is no information on the composition of collagen fibers in the fascial tissues of obese people compared to non-obese ones. Large-scale studies comprising more than 10,000 inguinal hernia patients demonstrated that the odds of groin hernia decrease with obesity relative to normal weight [34,35]. However, this association is not likely due to alterations in abdominal wall resistance. Rather, it is believed that the local accumulation of fat can somehow hold back the herniation process, e.g., by modifying the patency of the inguinal canal, or can make the defect less visible and consequently harder to diagnose [35].

No other factors potentially able to alter PINP and/or PIIINP results, such as osteoporosis, glucocorticoid use, or fibrogenic disorders, were present in our set of subjects; therefore, we can assume that hernia patients have constitutively lower PINP and higher PIIINP production.

Our data are partially discordant with those of Henriksen and colleagues [5] in that the PIVNP appeared unvaried in our patient cohort. Such discrepancy could depend on the small number of samples examined in both studies, but also on the different selection of control subjects, represented by healthy volunteers in our investigation and by patients undergoing cholecystectomy for gallstones in the previous one. Such a reference group requires caution in interpreting the results obtained, as alterations in collagen metabolism have been described in gallstones patients [36].

On the whole, our investigation further confirms that a dysregulation of collagen metabolism occurs in patients with inguinal hernia, recognizable by plasma measurements of peptides deriving from biosynthetic processes. The most relevant finding is the strong inverse correlation between inguinal hernia and PINP/PIIINP ratio, which represents an independent risk factor for disease. This parameter could be useful to distinguish individuals at higher risk of hernia development or recurrence, especially if subjected to additional risk factors such as weightlifting, abdominal surgery, or increased intra-abdominal pressure. A limitation of this work to assess the clinical relevance of PINP/PIIINP ratio is, beside the limited sample size, its case–control design, which entails the possibility of selection bias affecting the comparison between groups. A prospective research (randomized controlled or cohort study) is therefore warranted to further investigate the accuracy of PINP/PIIINP ratio as a biomarker of inguinal hernia. Moreover, it is essential to clarify the underlying causes of collagen imbalance at the tissue level: whether changes in the amount, activation status, and/or compartmentalization of enzymes involved in collagen metabolism occur, and whether the gene expression profile of type I and III collagens is altered in hernia patients. It would also be important to understand whether the causal factor(s) are congenital or acquired, in order to devise corrective and/or preventive therapies alternatives or complementary to surgery.

This research line will be relevant in the surgical field as well. In fact, if it were established that low PINP levels or PINP/PIIINP ratio are related to higher rate of recurrences and/or postoperative complications, this data would support the surgeon in choosing the most appropriate type of intervention and follow-up in individual cases. Improving the repair outcome or possibly avoiding groin hernia onset would be especially important in geographic areas where disease prevalence and burden are higher and there are greater barriers in accessing treatments, such as countries in the Global South [37].

Finally, this knowledge would usefully fit into the broader context of artificial intelligence (AI)-based personalized medicine. Advanced machine learning techniques applied to large amounts of multimodal data are currently able to make predictions on postoperative morbidity, mortality, and long-term outcomes, allowing for more tailored patient care [38], and it would be desirable to extend these practices also to hernia patients.

## 4. Materials and Methods

### 4.1. Study Participants

This study included 51 patients with inguinal hernia and 42 control subjects, enrolled according to the following inclusion criteria: male gender, age ≥ 18 years, absence of diseases potentially affecting collagen turnover (e.g., osteoporosis, glucocorticoid intake, fibrogenic disorders, cancer, inflammation, major injuries), absence of current or previous abdominal hernias and no family history of abdominal hernia in control subjects, and diagnosis of primary or recurrent inguinal hernia in patients. Plasma samples were collected from healthy volunteers and from patients admitted for surgical repair of inguinal hernia at the Department of Surgery of the University “Sapienza”, Umberto I General Hospital of Rome. Age, body mass index (BMI), and previous or current comorbidities of the participants were recorded. In addition, all participants answered a questionnaire regarding potential risk factors for abdominal hernia, i.e., disease familiarity, weightlifting for work or sport, tobacco smoking habits, constipation, and previous abdominal surgeries. Table 2 summarizes the information collected from this survey with which it was possible to run statistical analyses. Patients’ hernia characteristics are listed in Table 3.

The participants were monitored for their health status for at least 3 weeks prior to the blood sampling. They were also advised to maintain normal physical activity, avoid excessive physical efforts, and inform the doctor about any medications or supplements they were taking.

Blood drawings were made in the morning on fasting subjects and before performing the surgical procedure in the case of patients. Vacuum blood collection tubes containing heparin were pre-chilled on ice, and samples were centrifuged in a refrigerated benchtop centrifuge at 2000× *g* for 15 min to separate plasma from cellular component. Each plasma sample was aliquoted into sterile 1.5 mL pre-cooled tubes and aliquots were frozen at −80 °C within 30–60 min of collection.

### 4.2. Zymography Assay of MMPs

The gelatinases activity in plasma samples was evaluated as previously described [39]. Briefly, 0.1% gelatin substrate (Merck, Darmstadt, Germany) was embedded in 8% polyacrylamide gels. Three microliters of plasma were diluted 1 to 15 with non-reducing loading buffer and separated by sodium dodecyl sulphate–polyacrylamide gel electrophoresis. Serum-free conditioned culture medium of the human fibrosarcoma cell line HT1080 was used as positive controls. After the electrophoresis run, gels were washed and incubated overnight in a shaking bath at 37 °C with an appropriate buffer to activate MMPs. Negative controls were performed in parallel experiments by incubating gels in a calcium-free buffer containing 1,10-phenanthroline 10 mM, a reversible inhibitor of MMPs. Finally, gel images were digitized using the IBright FL1500 instrument (Thermo Fisher Scientific, Waltham, MA, USA), and densitometric analyses of the bands were performed with ImageJ software (Version 1.54p).

### 4.3. Magnetic Bead-Based Immunoassay of MMPs

Plasma samples of randomly selected controls (*n* = 10) and patients (*n* = 30) were used for quantitative determination of MMP-2 and MMP-9 proteins with the Luminex 200 System (Diasorin, Saluggia, Italy). The two analytes were simultaneously profiled with a customized Magnetic Luminex High Performance Assay kit (Bio-Techne, Abingdon, UK) according to the manufacturer’s instructions, and raw data were analyzed with the XPONENT management software of the instrument (Diasorin). All samples were measured in duplicate.

### 4.4. ELISA Assays

ELISA kits were used to estimate the plasma concentrations of LOX enzyme (MBS2704542 by MyBioSource, San Diego, CA, USA), PINP (E-EL-H0185 by Elabscience, Houston, TX, USA), PIIINP (E-EL-H0183 by Elabscience), and PIVNP (orb561684 by Biorbyt, Durham, NC, USA) following manufacturer instructions. All samples were read in duplicate in 96-well microplates with an 8-channel photometer Plate Reader (DAS, Rome, Italy). Four-parameter logistic standard curves were created with the online software package MyCurvefit and used to estimate sample concentrations by interpolating the absorbance values.

### 4.5. Statistics

Statistical analysis of data was performed by applying the non-parametric Mann–Whitney test to compare results of zymography and immunoenzymatic assays between controls and patients, smokers and non-smokers, and subjects with or without family history of hernia. When more than two categories were compared the non-parametric Kruskal–Wallis test was used. Correlation analyses between scalar variables, i.e., plasma concentrations of analytes, BMI, and age of the subjects, were performed by determining the Spearman rank correlation index R. Due to the low number of obese subjects in our dataset (5 controls and 5 patients), we pooled individuals with overweight (25 ≤ BMI < 30) or obesity (BMI ≥ 30). Logistic regressions were applied to estimate the odds ratios (ORs) associated with molecular and clinical factors. A ROC curve was created with PINP/PIIINP values, and the optimal cut-off was determined as the number that maximized the Youden index. The analyses were performed with SPSS version 27 software (Statistical Package for the Social Sciences, IBM), and statistical significance was fixed at the 5% level.

## Figures and Tables

**Figure 1 ijms-26-07032-f001:**
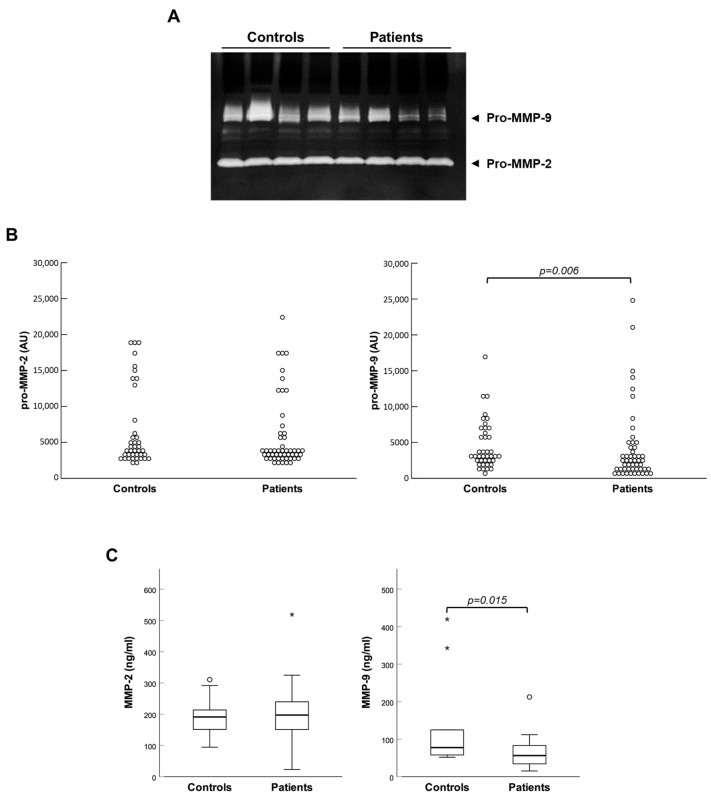
Plasma levels of gelatinases in controls and patients. (**A**) Representative image of zymography results with 4 controls and 4 patients samples; (**B**) scatter plots of the relative enzymatic activities of MMP-2 and MMP-9; (**C**) box plots of MMP-2 and MMP-9 plasma concentrations measured by magnetic bead-based immunoassay in subgroups of controls and patients. AU: arbitrary units; * extreme values.

**Figure 2 ijms-26-07032-f002:**
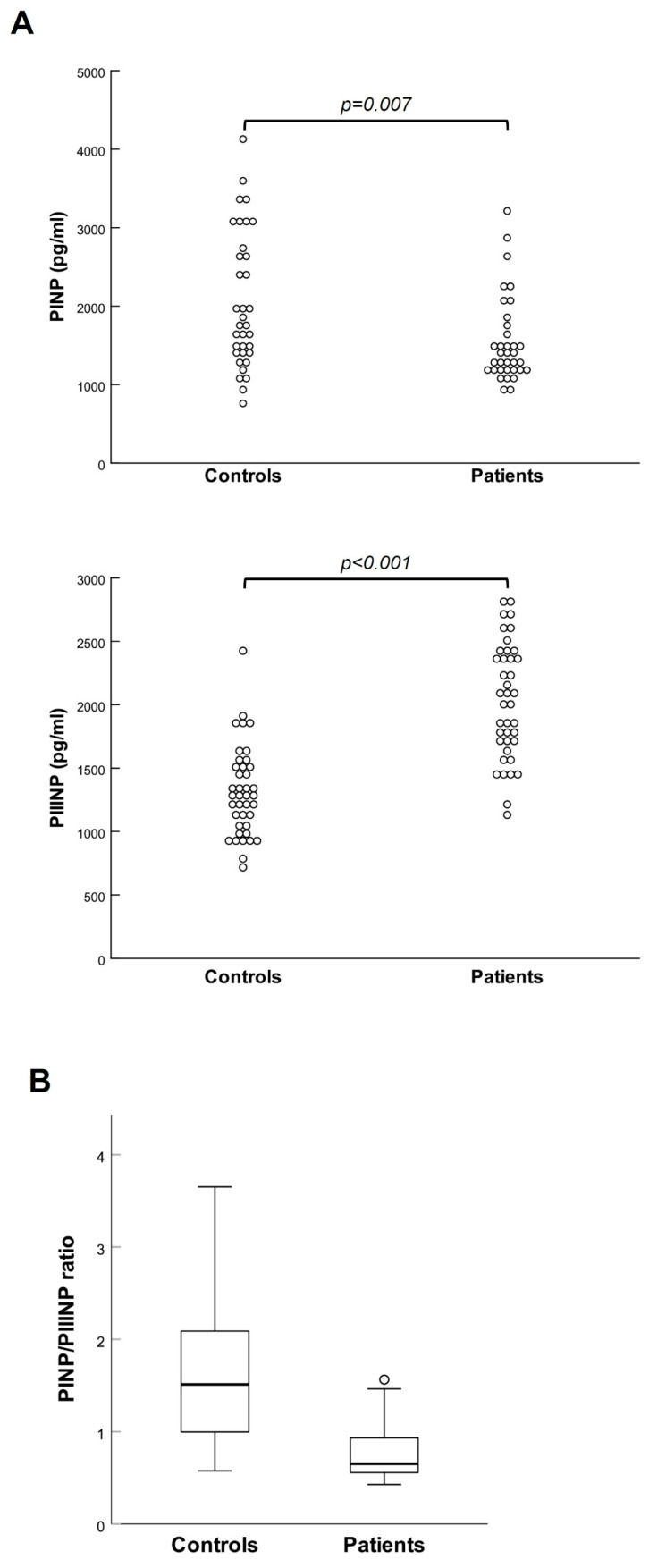
(**A**) Scatter plots of PINP and PIIINP plasma values in controls and patients; (**B**) box plot of the PINP/PIIINP ratio in controls and patients.

**Figure 3 ijms-26-07032-f003:**
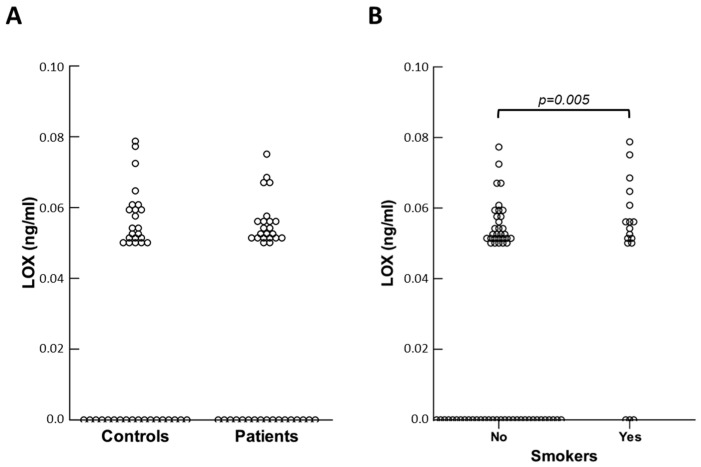
(**A**) Scatter plot of LOX plasma values in controls and patients; (**B**) scatter plot of LOX plasma values in smokers and non-smokers.

**Figure 4 ijms-26-07032-f004:**
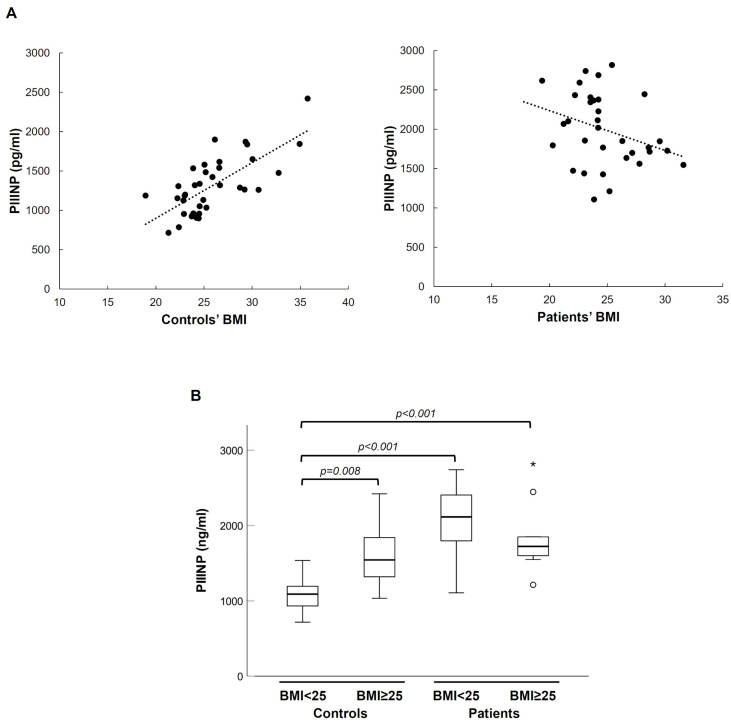
Correlation between circulating levels of PIIINP and BMI in controls and patients. (**A**) Scatter plot of PIIINP plasma concentrations and BMI in healthy subjects; (**B**) box plot of the distribution of PIIINP values in normal weight or overweight/obese controls and patients. * Extreme values.

**Figure 5 ijms-26-07032-f005:**
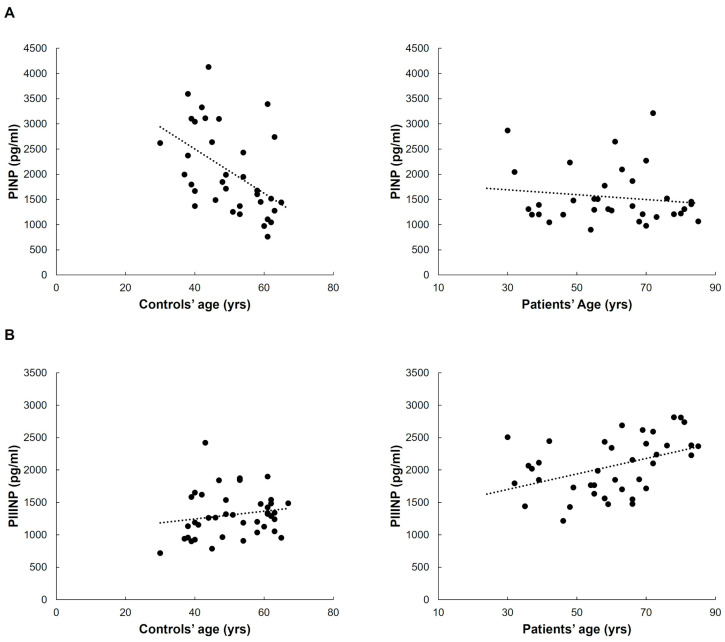
(**A**) Scatter plots of PINP plasma concentrations and ages of healthy subjects and patients; (**B**) scatter plots of PIIINP plasma concentrations and ages of healthy subjects and patients.

**Figure 6 ijms-26-07032-f006:**
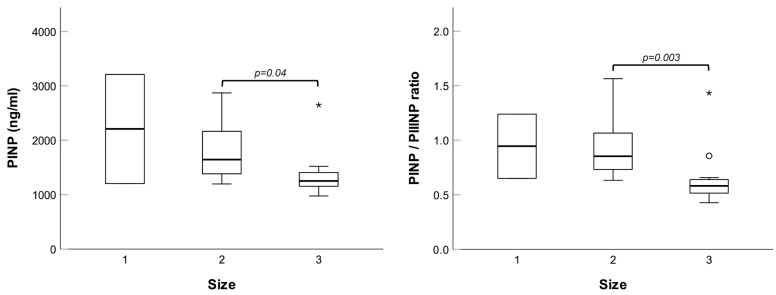
Box plots of PINP plasma concentrations and PINP/PIIINP ratio in patients with different hernia sizes. * Extreme values.

**Figure 7 ijms-26-07032-f007:**
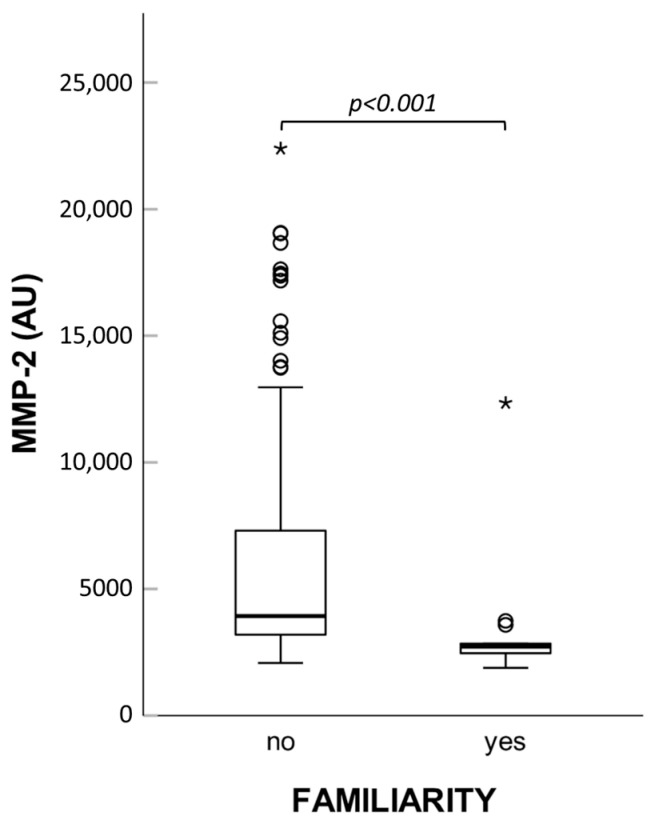
Box plot of MMP-2 enzymatic activity in plasma of patients with or without familial history of hernia. AU: arbitrary units; * extreme values.

**Figure 8 ijms-26-07032-f008:**
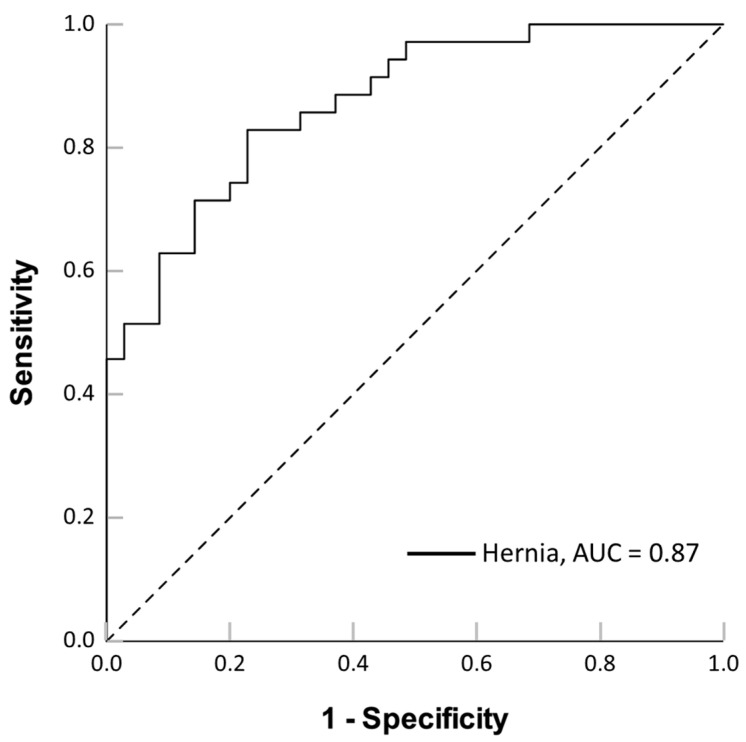
ROC curve for inguinal hernia prediction based on the PINP/PIIINP ratio.

**Table 1 ijms-26-07032-t001:** Logistic regression analysis with inguinal hernia as outcome variable and age, BMI, and PINP/PIIINP as explanatory variables. OR: odds ratio; CI: confidence interval.

	OR	95% CI	*p*-Value
Age	1.01	0.952–1.071	0.741
BMI	0.796	0.642–0.988	0.038
PINP/PIIINP	0.027	0.003–0.214	<0.001

**Table 2 ijms-26-07032-t002:** Characteristics of study participants.

**Age (Mean ± sd)**		
Controls	50.8 ± 10	
Patients	58.5 ± 16	
	**NO**	**YES**
**Weightlifting**		
Controls	28	14
Patients	39	12
**Smoke**		
Controls	35	7
Patients	38	13
**Benign prostate hypertrophy**		
Controls	36	6
Patients	41	10
**Obesity (BMI ≥ 30)**		
Controls	37	5
Patients	46	5
**Overweight/Obesity (BMI ≥ 25)**		
Controls	25	17
Patients	32	19
**Familial history**		
Controls	42	0
Patients	37	14

**Table 3 ijms-26-07032-t003:** Hernia types and sizes. n.a.: not available.

Hernia Features	Frequency
Size: 1	2/51 (3.9%)
2	12/51 (23.5%)
3	22/51 (43.1%)
n.a.	15/51 (29.4%)
Type: indirect	22/51 (43.1%)
direct	15/51 (29.4%)
bilateral	8/51 (15.7%)
recurrent	6/51 (11.8%)
n.a.	7/51 (13.7%)

## Data Availability

The data generated in the present study may be requested from the corresponding authors.

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
