# Peer review of "Circulating Biomarkers as Potential Risk Factors for Inguinal Hernia"

_ijms, 2025, doi:10.3390/ijms26157032_

Round 1

Reviewer 1 Report

Comments and Suggestions for Authors

WE would like to thank you for the submission of your work. There is a need to  the patient the risk to develop hernia and recurrence.

Here some comment: :

add the full name of MMPs, LOX,PINP, PIIINP etc

PINP, PIIINP can be add in the keywords

discussion of the clinical implications of those findings

Author Response

We are grateful to the Reviewer for his/her valuable work on this manuscript and for his/her appraisal. As suggested by the Reviewer, we modified the manuscript as follows:

  1. The full names of MMPs and LOX were added in the revised Abstract. However, having already exceeded the 200-word limit indicated by the journal, we would prefer not to add also “N-terminal propeptides of type I (PINP) and type III (PIIINP) procollagens” and “N-terminal propeptide of type IV pro-collagen (PIVNP)”. Actually, we used the wording “peptides produced during collagen biosynthesis (PINP, PIIINP, and PIVNP)” to explain more concisely the general meaning of the acronyms.
  2. We added “N-terminal propeptides of type I (PINP) and type III (PIIINP) procollagens” to the keywords.
  3. The Discussion was integrated with observations regarding the clinical implications of the presented findings.

Reviewer 2 Report

Comments and Suggestions for Authors

I have read the manuscript Circulating biomarkers as potential risk factors for inguinal hernia

which is very important because hernia is such a very common condition and presents a global burden. I commend the authors on this work we can perform surgery now on a molecular level and personally this what I think future is heading ( surgery at molecular level) many conditions are treated that way why not Hernia.

The introduction and discussion quite informative, surprisingly the methods came after discussions ideally comes before results , but may in this manuscript it helps the reader to spend all energy in understanding the main concept first before engaging in routine lab work.

Interesting that BMI is inverse to hernia formation, the molecular marker ratio as indicators as for hernia formation, the limitations are well described yes further studies are needed but the question which needs to be posed at end of the manuscript what next is reversing ration of those marker might improve the condition of the hernia ? Of course more studies needed more data needs to be collected and thanks to artificial intelligence this could be easily pooled nowadays

Please cite the following manuscripts

National Institute for Health and Care Research (NIHR) Global Health Research Unit on Global Surgery , Global access to technologies to support safe and effective inguinal hernia surgery: prospective, international cohort study, BJS, Volume 111, Issue 7, July 2024, znae164, https://doi.org/10.1093/bjs/znae164

And

Gumbs AA, Croner R, Abu-Hilal M, Bannone E, Ishizawa T, Spolverato G, Frigerio I, Siriwardena A, Messaoudi N. Surgomics and the Artificial intelligence, Radiomics, Genomics, Oncopathomics and Surgomics (AiRGOS) Project. Art Int Surg. 2023;3:180-5. http://dx.doi.org/10.20517/ais.2023.24

Author Response

We are grateful to the Reviewer for his/her valuable work on this manuscript and for his/her appraisal.

The Discussion has been integrated with some considerations on the clinical implications of the presented results and on future research perspectives, and we have added the suggested references (38 and 39 of the revised manuscript).

Reviewer 3 Report

Comments and Suggestions for Authors

This article studies the PINP/PIIINP ratio as potential risk factors for inguinal hernia, which has certain clinical significance. These studies are very common in tumors, but are rare in benign inguinal hernia, with a small number and a lack of large-sample studies. Table2 is not a standard academic table. You can refer to the attachment I provided.

Author Response

We are grateful to the Reviewer for his/her valuable work on this manuscript and for his/her appraisal.

We have modified Table 2 hoping to have interpreted the suggested changes correctly.